# Sustainability-inspired cell design for a fully recyclable sodium ion battery

Tiefeng Liu[1], Yaping Zhang[2], Chao Chen[1], Zhan Lin[1], Shanqing Zhang[3] & Jun Lu [4]

Large-scale applications of rechargeable batteries consume nonrenewable resources and produce massive amounts of end-of-life wastes, which raise sustainability concerns in terms of manufacturing, environmental, and ecological costs. Therefore, the recyclability and sustainability of a battery should be considered at the design stage by using naturally abundant resources and recyclable battery technology. Herein, we design a fully recyclable rechargeable sodium ion battery with bipolar electrode structure using $Na_3V_2(PO_4)_3$ as an electrode material and aluminum foil as the shared current collector. Such a design allows exceptional sodium ion battery performance in terms of high-power correspondence and long-term stability and enables the recycling of ~100% $Na_3V_2(PO_4)_3$ and ~99.1% elemental aluminum without the release of toxic wastes, resulting in a solid-component recycling efficiency of >98.0%. The successful incorporation of sustainability into battery design suggests that closed-loop recycling and the reutilization of battery materials can be achieved in next-generation energy storage technologies.

---

[1] School of Chemical Engineering and Light Industry, Guangdong University of Technology, 510006 Guangzhou, China. [2] State Key Laboratory of Environment-friendly Energy Materials, School of Materials Science and Engineering, Southwest University of Science and Technology, 621010 Mianyang, China. [3] Centre for Clean Environment and Energy, Environmental Futures Research Institute and Griffith School of Environment, Griffith University, Gold Coast Campus, Brisbane, QLD 4222, Australia. [4] Chemical Sciences and Engineering Division, Argonne National Laboratory, 9700 Cass Ave, Lemont, IL 60439, USA. Correspondence and requests for materials should be addressed to Z.L. (email: zhanlin@gdut.edu.cn) or to S.Z. (email: s.zhang@griffith.edu.au) or to J.L. (email: junlu@anl.gov)

Rechargeable batteries have powered extensive types of mobile electronics and shown extraordinary promise for grid storage[1–3]. However, these potential applications call for extremely large-scale battery production and consequently cause growing concerns regarding the massive quantities of end-of-life batteries that seriously threaten environmental and ecological safety[4]. The recycling of end-of-life batteries not only minimizes the demand for critical material resources but also addresses significant concerns regarding environmental pollution and ecological impacts[5]. Generally, a typical battery is assembled with six components: accessories, electrolyte, separator, cathode, anode, and current collectors[6]. Although accessories, electrolytes, and separators have been efficiently recycled[5], it is a substantial challenge to recover the electrode unit containing the current collectors, anode, and cathode, which are the most economically valuable components. The lithium-ion battery (LIB), for example, adopts a unipolar electrode structure[6], where cathode materials and anode materials are coated on Al and Cu foils, respectively (shown as one unit in Fig. 1a). In the course of large-scale recycling of LIB units, cathode materials, anode materials, and current correctors will inevitably be mixed together, leading to difficult separation where recycling is not economically viable[4]. Hydro- and pyrometallurgical recovery of valuable metals in LIBs has been widely used, providing a possible route for decreasing the demand for mining and enhancing sustainability. These metals can be efficiently recycled, purified, and reused in new battery production. However, there are obvious disadvantages in terms of the high cost and substantial production of nonecologically friendly solid, liquid, and gas wastes[5]. The potential grid-level energy storage of electricity generated from intermittent renewable sources (e.g., solar and wind) will require vast numbers of LIB cells, thereby creating an even greater drain on rare metal sources (such as lithium) and more extensively impacting the environment when end-of-life LIB cells cannot be effectively recycled.

To address these issues, we believe that the recyclability and sustainability of a battery must be considered at the design stage[4]. Sodium battery technology could be a promising alternative to LIBs for grid-level energy storage due to the widely established competitive energy and power densities, low cost, and environmental benignity of sodium batteries[1,7,8]. One good example is high-temperature sodium–sulfur batteries (HT-NSBs), which were successfully commercialized in the 1980s. Furthermore, sodium salts are highly available[9]. Recently, room-temperature sodium-ion batteries (NIBs) have been considered a safer energy storage technology than HT-NSBs. However, NIBs have a similar electrode structure to that of LIBs[8] and thereby experiences the same recovery issues when applied to grid-level energy storage. The historical crisis of white plastics[10] should teach us that prescient product design could help increase recycling and reduce environmental impact.

Here, to achieve viable recyclability and sustainability of NIBs, we design a bipolar electrode structure instead of a unipolar electrode structure: the cathode and anode materials are coated on both sides of a shared metallic current collector (Fig. 1b). Al foil and $Na_3V_2(PO_4)_3$ (NVP) are selected for the shared current collector and active materials of the cathode and anode, respectively. Al foil enables the bipolar electrode structure to benefit from the lack of alloying reaction between Na and Al. The electrode materials are readily collected from waste NIBs via a basic treatment. Consequently, over 98.0% of the solid components of waste NIBs are recycled without the release of toxic wastes. In particular, the high-value NVP is separated and collected with a recycling efficiency of 100%. Recycled NVP simply treated by a direct regeneration process is reused in a new NIB and exhibits excellent electrochemical performance. Therefore, the adoption of

such a bipolar electrode structure for NIBs confers the NVP material with closed-loop sustainability.

## Results

**A fully recyclable sodium ion battery design.** In traditional NIBs, Al and Cu foils are used as current collectors for the cathode and anode, respectively. Their electrochemical applicability as a universal current collector was investigated in the full voltage window of 0–5 V, as shown in Fig. 1c. Al foil shows a negligible current from 0 to 5 V with high stability, while Cu foil exhibits a sharp rise in current from ca. 3.6 V due to the oxidation of Cu at high potentials[11]. The discharge platform of Al foil is under 0 V, as shown in Fig. 1d, demonstrating that there is no alloying reaction between Al foil and metallic Na. This is further confirmed by the existence of metallic Na in ex situ XRD (Supplementary Fig. 1). Therefore, we designed an NIB with a bipolar electrode structure using Al foil as a single current collector, simplifying the electrode structure to achieve more efficient recycling and saving costs by eliminating the need for a Cu foil current collector.

After designing the bipolar electrode structure and confirming the selection of a single Al foil current collector, we examined active materials by establishing a system among the three with water to form NaOH (Eq. 1) and then corrode the Al foil (Eq. 2). Elemental Al can be collected in the form of $Al(OH)_3$ via acid-base neutralization with a suitable amount of HCl (Eq. 3). The final waste is mainly NaCl solution. In this work, we chose NVP as the cathode material due to its high rate performance, long lifespan, and abundant reserves[12–14]. Highly toxic elemental V needs to be recycled[15]. Consequently, the combination of serial chemical reactions (i.e., Eqs. 1–3) among the cathode, anode, and current collector not only facilitates the rapid separation of individual cell components but also enables the effective recycling of specific materials. To recycle the electrolyte and stainless steel accessories, NIBs can share the recycling technologies of LIBs[5,16].

$$Na(s) + H_2O(l) \rightarrow NaOH(aq) + H_2(g) \tag{1}$$

$$NaOH(aq) + Al(s) \rightarrow NaAlO_2(aq) + H_2(g) \tag{2}$$

$$NaAlO_2(aq) + HCl(aq) \rightarrow NaCl(aq) + Al(OH)_3(s) \tag{3}$$

**A high-performance and stable sodium ion battery.** Based on the above design, we assembled an asymmetric NIB cell with two units (Supplementary Fig. 2). The cell consists of six components: (i) carbon-coated NVP (NVP@C) cathode (Supplementary Fig. 3); (ii) metallic Na anode; (iii) Al foil current collector; (iv) 1M $NaClO_4$ in ethylene carbonate (EC) and diethylene carbonate (DEC) as the electrolyte; (v) glass fiber separator; and (vi) accessories including stainless steel cases and seal rings. Fig. 2a shows the initial charge and discharge curves of a two-unit NIB cell. Benefitting from the voltage contribution of each unit (~3.4 V per unit), the NIB cell readily exhibits high voltage platforms of ca. 6.9 V and 6.5 V at the charge and discharge states, respectively. Stable cycling of over 100 times demonstrates the effectiveness of the bipolar electrode structure (Fig. 2b). Further illustrating the high-rate features of the bipolar electrode structure, the NIB cell exhibits remarkable rate performance, i.e., up to 20 C, with a high discharge voltage (Fig. 2c). The bipolar NIB cells yield an energy density of ca. 368.2 Wh kg$^{-1}$ (normalized to NVP@C, i.e., 92.1 Wh kg$^{-1}$ when normalized to the full cell) at 1 C and a power density of ca. 7.3 kW kg$^{-1}$ (normalized to NVP@C, i.e., 1.83 kW kg$^{-1}$ when normalized to the full cell) at 20 C[17], which make them potentially applicable in grid-level energy storage. Peak regulation in grid-level energy storage for

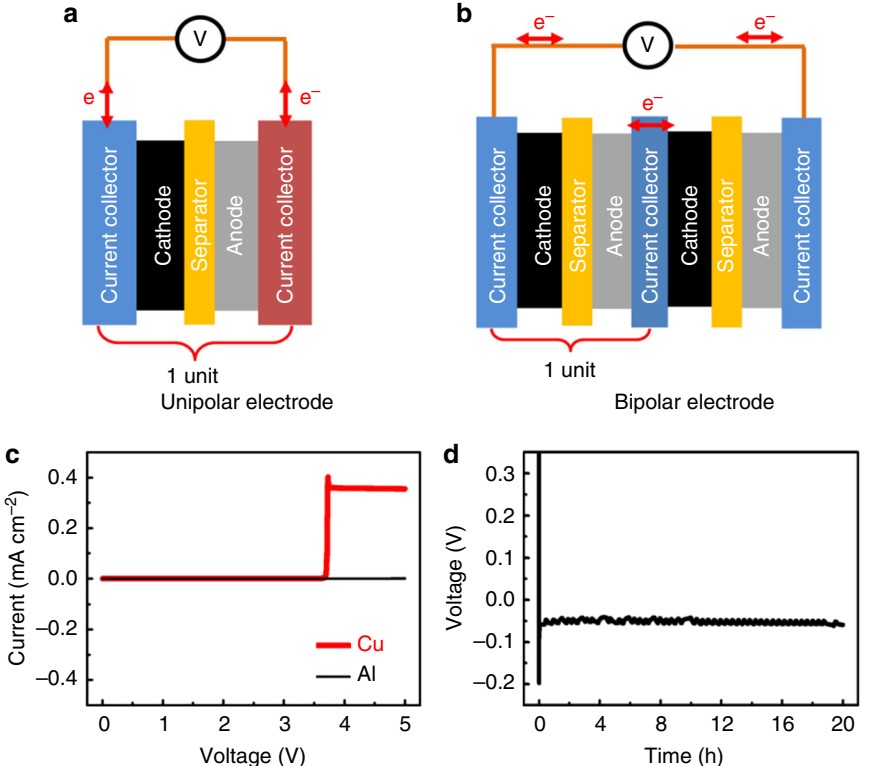

**Fig. 1** Design and achievement of bipolar electrode structure. **a** Schematic of the conventional unipolar electrode structure (one-unit cell). **b** Schematic of the proposed bipolar electrode structure (two-unit cell). **c** Linear scanning voltammograms of Al and Cu foils as the cathodes with metallic Na as the anode at 0~5 V at a scanning speed of 0.2 mVs$^{-1}$. **d** Discharge curves of Al foil as the cathode with metallic Na as the anode at 100 mA cm$^{-2}$ for 20 h

intermittent renewable sources—which require corresponding high-power batteries—is a typical application[1,7]. To simulate this application, pulse current charge/discharge tests were performed. At a rate of 20 C for 30 s, the voltage platforms during charge/discharge are relatively steady within the pulse period (Supplementary Fig. 4). The large cross-sectional area between the cathode and anode enables low and homogeneous current density, resulting in small voltage drops even at high rates. The NIB cell repeated cycling over 2000 times (Fig. 2d), illustrating its remarkable stability.

We can further enlarge the energy and power densities of the NIB by assembling battery packs from two to three (Supplementary Fig. 5a) and five (Supplementary Fig. 6a) units. The output voltage of the cells linearly increases with the number of units. For three units, the NIB cell delivers a discharge voltage of ca. 9.8 V at 10 C with high power and excellent cycling (Supplementary Fig. 5b and 5c). The discharge voltage is increased to 16.4 V for five units with long-term cyclability (Supplementary Fig. 6b and c). The high performance is attributed to the proposed NIB design combining a bipolar electrode structure with a high-rate electrode material.

**Fully rejuvenated electrochemical performance of the sodium ion battery**. The most important advantage of our designed NIB with a bipolar electrode structure is the recyclability of the electrode materials in the battery unit, i.e., NVP and elemental Al. After cycling, we disassembled a two-unit NIB cell to investigate the feasibility of the proposed recyclability. (Supplementary Movie 1 shows detailed operations and phenomena in the recycling process). The NIB cell was first predischarged and subsequently disassembled into fractions. The electrolyte was difficult to recycle due to the small amount of organic electrolyte used in

coin-type NIB cells. In the scope of this work, we focus on solid-component recycling in spent NIBs with a bipolar electrode structure. All fractions were immersed in a recycling tank filled with water at room temperature. The NVP@C slurry was naturally detached from the Al foil (Supplementary Fig. 7), and hydrogen was the byproduct (Eqs. 1, 2). After filtration, we easily recovered the stainless steel cases, sealing rings, and glass fibers; the cathode slurry containing NVP@C, the conductive additive, and the polymeric binder was also collected.

An additional washing step for the recycled NVP slurry was performed, leading to a 20% increase in wastewater. This wastewater from additional washing steps can be reused in recycling tanks. The residual contamination was tested by ICP. The vanadium and phosphate concentrations were less than 2 ppb in the leachate, indicating that the NVP@C is very stable in basic solution. In addition, no Al residual was detected in the recycled NVP sample. Subsequently, according to the TGA curve of NVP@C, the carbon coating on the surface of the NVP particles was burned at 500 °C, while NVP was oxidized by V from +3 to +5 as the temperature increased (Supplementary Fig. 8). Therefore, the obtained cathode slurry was sintered in air at 550 °C to recycle NVP. Appropriate amounts of diluted HCl solution were used to collect elemental Al in the form of Al(OH)$_3$, with mainly NaCl aqueous solution left (Eq. 3).

Weight breakdowns of the NIB cell in the initial and recycled states are listed in Supplementary Table 1 and Fig. 9. In addition to achieving almost 100% recycling of accessories and separators, we successfully recycled almost 100% of the NVP from the cathode slurry and 99.1% of the elemental Al (Supplementary Fig. 10 and Note 1). Compared with hydro- and pyrometallurgical recovery of valuable metals[18], our process for solid-component recycling is cost-saving, highly efficient, and environmentally friendly[4]. The whole solid-component recycling

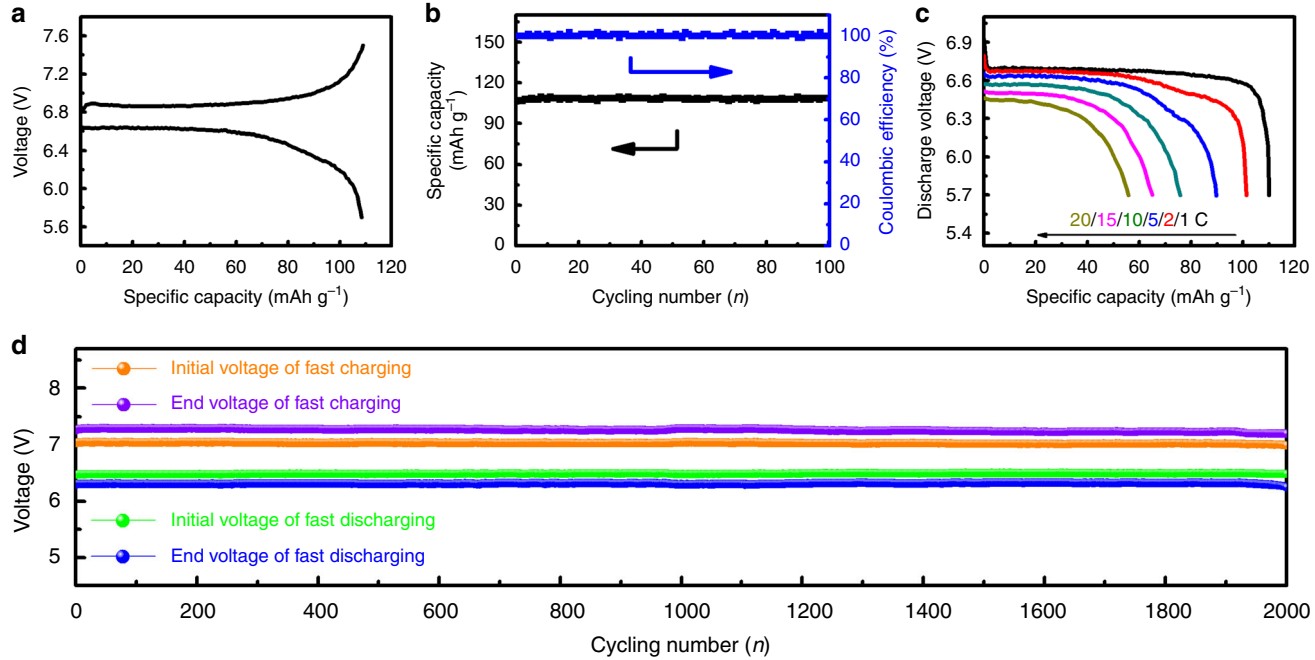

**Fig. 2** Electrochemical behavior of a two-unit bipolar NIB cell. **a** Initial charge and discharge curves at 1 C in a voltage range from 5.7 to 7.6 V. **b** Cycling 100 times at 1 C. **c** Rate capabilities from 1 to 20 C. **d** The voltages of long cycles at the beginning and the end of each step under fast charge/discharge

efficiency of our proposed NIB cell is ca. 98.1%, which is close to that of lead-acid batteries with heavily regulated disposal[4]. The only losses are the metallic Na, conductive additive, and polymer binder (Supplementary Fig. 11).

More importantly, this design can also be extended to other battery systems. As an example, we also assembled a symmetric NIB unit (Supplementary Fig. 12a) where NVP acts as the active material in both the anode and cathode in a symmetric configuration (Supplementary Note 2). The symmetric NIB cell also exhibits stable electrochemical performance (Supplementary Fig. 12b). Because both the cathode and anode use identical NVP@C material, diluted NaOH solution instead of water was used to separate the NVP and recycle elemental Al in the recycling process. The recovery for this symmetric NIB cell (Supplementary Movie 2) is therefore as convenient as that for the asymmetric NIB cell mentioned above. A solid-component recycling efficiency of ca. 99.7% was achieved with this system (Supplementary Fig. 13 and Table 3).

The cathode is the most expensive part of a battery cell and, to a great extent, determines the cost of the cell[17,19]. After recycling, we continued to demonstrate the sustainability of recycled NVP in a closed loop, as shown in Fig. 3. The most efficient and simple recovery method is to rejuvenate the cathode, which is called direct recovery[4]. In other words, cathode materials can be recycled and reused in the new battery design with a simple and low-cost method rather than conventional hydro- or pyrometallurgic processes, significantly saving the environmental and material costs of the fundamental elements of the cathode. We first reprocessed the recycled NVP for a new battery unit by direct regeneration[20–22]. The recycled NVP was calcined at 800 °C in an inert atmosphere with appropriate amounts of sucrose to form reprocessed NVP@C. The XRD patterns (Fig. 4a) are well indexed to a high-crystallinity NASICON structure (JCPDS No. 62-0345) without impurity, indicating there is no decomposition of the recycled NVP[12].

The scanning electron microscopy (SEM) image (Supplementary Fig. 14a) demonstrates that the morphology (including particle size and porosity) of reprocessed NVP@C is similar to

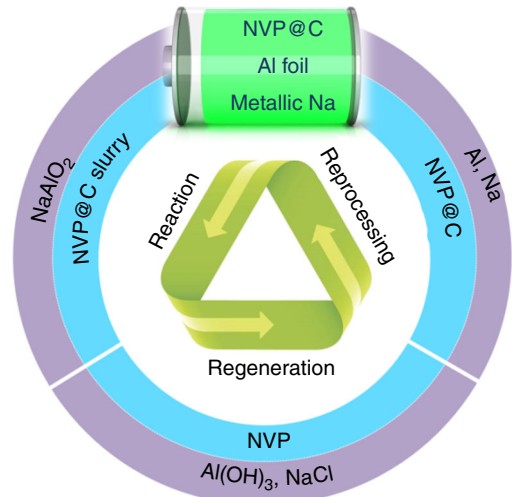

**Fig. 3** Schematic closed-loop utilization of a proposed NIB unit: Loop 1 (blue), reformation of NVP@C from the recycled NVP; Loop 2 (purple), recycling of elemental Al in the form of Al(OH)$_3$ and elemental Na in the form of NaCl through the processes of reaction, regeneration, and reprocessing

that of the original NVP@C (Supplementary Fig. 14b). The electrochemical behavior (Fig. 4b) and cyclic voltammetry measurement (Fig. 4c) of reprocessed NVP@C are similar to those of the original NVP@C, demonstrating that the reprocessed NVP@C has good electrochemical reversibility. The reprocessed NVP@C material is also capable of fast charging and discharging (Supplementary Fig. 15a and 15b). The long cycling performance of 500 cycles further confirms the excellent capacity retention of reprocessed NVP@C (Supplementary Fig. 15c). Furthermore, the NVP recycled from the above-tested NIB cells was repeatedly implemented. The reprocessed NVP still exhibited very good performance with stable long-term cycling in NIBs

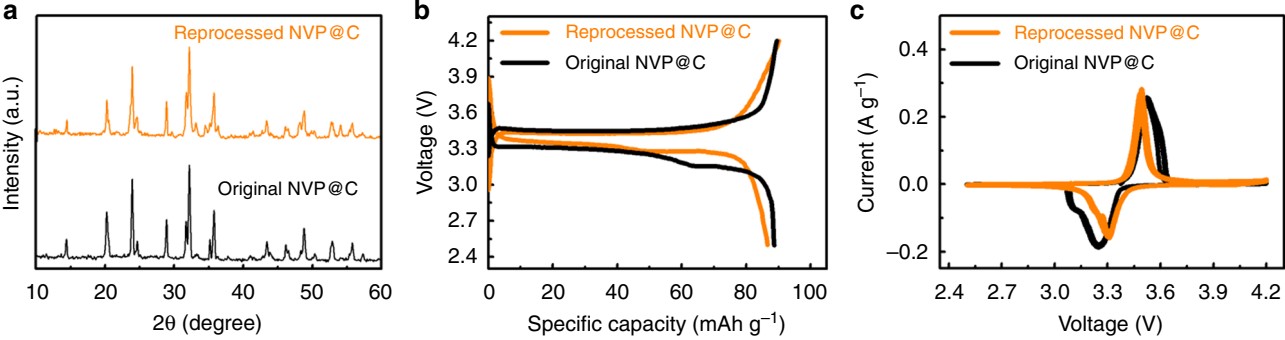

**Fig. 4** Characterizations of the reprocessed NVP@C compared with the original NVP@C. **a** XRD spectra. **b** Charge/discharge curves at 5 C. **c** Cyclic voltammetry curves

(Supplementary Fig. 16). All of the above results demonstrate the closed-loop utilization of the NVP material for a sustainable NIB with a bipolar electrode structure.

## Discussion

We attribute the above-demonstrated exceptional rate performance and full recycling operation of NIBs to the unique design of their bipolar electrode architecture (Fig. 1b). First, a bipolar electrode structure naturally accomplishes an internal series sequence without additional connecting accessories, allowing high current density transfer between the cathode and anode. Second, Al foil as a universal current collector for both the cathode and anode in the NIB enables the achievement of the proposed bipolar electrode structure and simplifies the NIB components. Third, the basic chemicals help establish an association among Al, metallic sodium and water, not only facilitating the separation of the active material (i.e., NVP) from the current collector but also enabling the effective recycling of Al, producing mainly NaCl aqueous solution as waste. Fourth, the NVP is directly regenerated without complex synthetic processing, and the bipolar electrode structure enables the rechargeable battery to use a low amount of steel and Al, which favors decreased energy consumption in the recycling process. Finally, Fig. 4 shows the fully rejuvenated electrochemical performance of the NVP in NIBs, demonstrating our proposed battery concept of performance, recyclability, and reuse, which critically depends on the initial stage of battery design. Although NVP reprocessing was only performed twice due to time constraints, we believe that recycled NVP can be further processed many times, and the current work has demonstrated the feasibility of closed-loop recycling of NVP.

Further explanation is needed here to claim that we have developed a fully recyclable NIB. First, the electrolyte can be recovered by sharing the LIB technologies. In general, electrolyte salts are dissolved in organic solvents (e.g., propylene carbonate and ethylene carbonate) with functional additives (e.g., vinyl carbonate and biphenyl). Currently, the main electrolyte recycling methods are classified as vacuum pyrolysis, organic solvent extraction, and supercritical $CO_2$ extraction[23]. Referring to a previous work by Liu et al., high-efficiency recycling of the electrolyte has been demonstrated, and the recycled electrolyte can be reused in new batteries[16,24]. Second, the recycling of metallic anodes is also highly feasible. Currently, metallic Na and Al are mainly produced by the electrolysis of molten salts at high temperature. We can expect that metallic Na and Al can be extracted well by melting recycled NaCl and $Al(OH)_3$ at 600 and 950 °C through an electrolysis process[25] for further usage, respectively. Although these recycling and reprocessing steps for the electrolyte and metallic Na/Al sources were not performed in this work, we think the above-verified knowledge enables a fully recyclable NIB.

In addition, safety is the first priority during practical recycling processes and emission treatment. On the one hand, hydrogen as a byproduct may pose safety concerns in recycling processes. Therefore, special treatment must be performed prior to battery recycling. For waste NIBs using metallic Na, the remaining metallic Na is highly active. The waste NIB is first cut into small pieces. The resultant slags should be placed in a dry ventilated place. The Na can react with $O_2$ to form $Na_2O$ (Eq. 4), and subsequently, some of the $Na_2O$ can react with $CO_2$ further to form $Na_2CO_3$ (Eq. 5). If this operation is executed well, the amount of byproduct hydrogen can be significantly reduced with the addition of $H_2O$ into the NIB waste (Eq. 6) compared with the amount obtained from direct water treatment. Moreover, the Al foil can be collected prior to full dissolution. We found that the surface of Al foil is readily corroded by alkali solution, leading to rapid separation of the electrode slurry and Al foil (Supplementary Fig. 7). Therefore, the fast recycling of Al foil delaminated from the electrode slurry can reduce the amount of hydrogen in the recycling process. Notably, ventilation must be carried out from the beginning to the end of the battery recycling process. On the other hand, the waste gas should be considered with regard to safety issues involving the environment and humans. The fluoridated binder (PVDF) is first directly burned to remove the binder from the recycled NVP@C slurry. However, this process can produce volatile organic compounds (VOCs), which are highly toxic to the environment and humans. The off-gas must be treated further at higher temperature to enable the efficient decontamination of VOCs[26].

$$Na(s) + O_2(g) \rightarrow Na_2O(s) \tag{4}$$

$$Na_2O(s) + CO_2(s) \rightarrow Na_2CO_3(s) \tag{5}$$

$$Na_2O(s) + H_2O(l) \rightarrow NaOH(aq) \tag{6}$$

In summary, our proposed bipolar electrode configuration for NIBs achieves high rate performance and full recycling operation. By taking advantage of this electrode design, we significantly diminish battery cost with reduced consumption of energy and material resources, making the NIBs environmentally sustainable for grid-level energy storage, in contrast to several traditional recycling methods (Supplementary Table 2). This design presents a possible solution to the most recent concerns regarding abandoned batteries with low recycling efficiency and high environmental risk[4,27]. In the future, some features are a must for NIBs used in large-scale energy storage systems: (1) A high recycling rate, enabling a fast response to energy fluctuations in renewable energy and peak regulation in smart grids. (2) A low battery cost, making batteries more economical by reducing the consumption of energy and material resources. (3) Excellent environmental

sustainability, diminishing all the traditional environmental risks from battery production, abandoned batteries, battery recycling and so on. We believe that a comprehensive consideration of recyclability and sustainability in battery design and material selection could make next-generation energy storage systems highly efficient and fully green.

## Methods

**Synthesis of NVP@C composite**. The synthesis of carbon-coated $Na_3V_2(PO_4)_3$ (NVP@C) composite was detailed in our previous work[12]. Briefly, $NH_4H_2PO_4$ (AR), $NH_4VO_3$ (AR), $Na_2CO_3$ (AR) and sucrose were weighed in a molar ratio of Na:V:P:C = 3:2:3:4. All raw materials were mixed, dispersed in 30 ml alcohol and ball-milled for 6 h in a planetary mill at 400RPM. The as-obtained mixture was dried at 80 °C in an oven to evaporate the alcohol. The remaining precursor was transferred into a porcelain boat and calcined at 300 °C for 2 h to obtain the bulk intermediate, which was ball-milled again for 6 h in alcohol and dried overnight. The resultant nanosized intermediate was sintered at 800 °C for 8 h under an Ar atmosphere in a tube furnace and cooled to room temperature to yield NVP@C.

**Material characterization**. The morphology and structure of the samples were investigated by a field-emission scanning electron microscope (SEM) on SU8010. X-ray diffraction (XRD) patterns were recorded on a Rigaku D/MAX-2550-PC X-ray diffractometer with Cu Kα radiation (λ = 0.154 nm). The 2θ degree range used in the measurements was from 20° to 80°. Thermogravimetric analysis (TGA) was performed on a Pyris 1 TGA (Perkin Elmer) system under air flow (50–800 °C, 10 °C min$^{-1}$). Inductive coupled plasma (ICP) emission spectrometry was executed on an iCAP6300 (detection limit: 2–200 μgL$^{-1}$).

**Assembly of NIB cells**. NVP@C (80 wt. %), acetylene black (AB, 10 wt. %), and polyvinylidene difluoride (PVDF, Aldrich) binder (10 wt. %) were dispersed in N-methyl-2-pyrrolidone (NMP, Aldrich) to form a homogeneous slurry. The obtained slurry was spread on an Al foil and dried at 100 °C overnight in a vacuum oven to remove the NMP solvent. The working electrode was round in shape with a diameter of 12 mm. The average loading mass of the NVP@C electrode was 2–5 mg cm$^{-2}$. The cell was assembled based on an asymmetric configuration, meaning that the cathode and anode used different active materials. An asymmetric bipolar NIB cell was fabricated with an NVP@C electrode, metallic Na disk, glass fiber (GA-55) separator, sealing ring, and organic electrolyte in an Ar-filled glove box. $NaClO_4$ (1 M) in a 1:1 (v/v) mixture of ethylene carbonate (EC) and propylene carbonate (DEC) was used as the electrolyte, which was completely sealed by the Al current collector and sealing ring in each independent cell. In addition, the cells were assembled based on a symmetric configuration for comparison, meaning that the cathode and anode used the same active material. A symmetric bipolar NIB cell was assembled with an NVP@C cathode, NVP@C anode, glass fiber (GA-55) separator, sealing ring, and organic electrolyte in an Ar-filled glove box. The organic electrolyte and sealing conditions were the same as those for the asymmetric bipolar NIB cell mentioned above.

**Electrochemical test**. For the NIB cells with bipolar electrode structure, galvanostatic charge-discharge tests were conducted using a custom-made LANDCT battery tester at different rates. Fast charge and discharge tests, which simulate an electrochemical battery system with high-power input/output to stabilize grid fluctuations arising from intermittent wind or solar power supplies[1], were performed over different times and current densities in a Solartron Analytical Cell Test System (1470E multichannel potentiostats). During the fast charge and discharge tests, the initial voltage of fast charging, end voltage of fast charging, initial voltage of fast discharging, and end voltage of fast discharging were recorded in each cycle (Supplementary Fig. 5). For recycled NVP@C–Na cells, galvanostatic charge–discharge tests were investigated using LANDCT 2001A battery testers at different rates with a voltage window of 2.5–4.3 V. Cyclic voltammetry measurements (2.5–4.3 V, 0.1 mVs$^{-1}$) were performed on a CHI 660E electrochemical workstation. For the characterization of Al and Cu foils, linear sweep voltammetry (LSV, 0.01–5.0 V, 0.2 mVs$^{-1}$) on a CHI 604E electrochemical workstation was used. For NVP@C–NVP@C cells, galvanostatic charge–discharge tests were conducted using LANDCT 2001A battery testers at different rates with a voltage window of 2–4.4 V. All the voltages are referenced to Na/Na$^+$.

**Reaction, regeneration, and reprocessing procedure**. The NIB cell containing the metallic Na anode was disassembled and soaked in water for 1 h. During this period, the Al foil current collector was dissolved in the NaOH solution formed from the reaction between metallic Na and water. After filtering the solution, the stainless steel cases, sealing rings, and separator were collected. The NVP@C slurry was also collected and dried overnight (Supplementary Fig. 8). The recycled NVP@C slurry was sintered at 550 °C for 4 h in air to remove the coating carbon, conductive additive, and polymer binder[28] and then cooled to room temperature to produce NVP powder. The recycled NVP powder was mixed with sucrose and further sintered at 800 °C for 6 h in an inert atmosphere to obtain reprocessed

NVP@C. Sucrose was weighed at a molar ratio of V:C = 1:2. For the NIB cell using NVP@C as the anode, the recycling process was the same except that diluted NaOH solution was used instead of water. The Al was recycled using an appropriate amount of diluted HCl solution to deposit white $Al(OH)_3$. The pH value of the solution was tuned to ca. 5.37. The recycling, reprocessing, and reuse of the NVP@C materials in bipolar NIBs was performed two times. Each time, all NVP@C was reused in NIBs after regeneration.

**Calculation of energy and power densities**. The energy density of NVP was calculated according to Eq. 7:

$$W = U \times Q \qquad (7)$$

where $W$ represents the specific energy density of NVP, $U$ represents the terminal voltage of one cell, and $Q$ represents the specific capacity of NVP at different rates. The power density of NVP was calculated according to Eq. 8:

$$P = U \times I \qquad (8)$$

where $P$ represents the specific energy density of NVP, $U$ represents the terminal voltage of one cell, and $I$ represents the overall current of NVP. The energy and power densities of the cell were calculated by referring to the weight percent of cathode material[17,29], which accounts for one quarter of the weight of a full NIB. Therefore, one full cell theoretically has 25% energy and power densities based on NVP.

## Data availability

The data that support the findings of this study are available from the corresponding authors upon reasonable request.

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

## Acknowledgements

Z.L. gratefully acknowledges the funding support from the National Natural Science Foundation of China (Project 51874104) and Key Technology and Supporting Platform of Genetic Engineering of Materials under State's Key Project of Research and Development Plan (Project 2016YFB0700600). S.Z. acknowledges the financial support from the Australia Research Council and Griffith University. J. L. gratefully acknowledges support from the U. S. Department of Energy (DO E), Office of Energy Efficiency and Renewable Energy, Vehicle Technologies Office. Argonne National Laboratory is operated for the DOE Office of Science by UC hicago Argonne, LLC, under contract number DE-AC02-06CH11357.

## Author contributions

T.L., Z.L., S.Z. and J.L. conceived of the idea. T.L.,Y.Z. and C.C. performed the material fabrication and electrochemical tests. T.L., Z. L., S.Z. and J.L. cowrote the paper. All the authors discussed the results and commented or revised the manuscript.

## Additional information

**Competing interests:** The authors declare no competing interests.

