## [Peer Review File · Nature Communications]

Reviewers' comments:

Reviewer #1 (Remarks to the Author):

In this manuscript, Lu and Lin et al. reported a novel design concept of bipolar sodium-ion battery (SIB) for large-scale energy storage with the potential to significantly reduce the cost of electrode materials due to the viability of high-efficiency recycling. The usage of light-weight and low-cost Al current collect may also bring benefits, like further increase the system energy density, reducing the cell cost etc. Another important feature is the capability of direct regeneration of the NVP cathode material, which is significantly different from the conventional pyro- and hydrometallurgical recycling processes. The potential advantages of direct regeneration without fully destructing the active materials include reduced operation cost (process, energy, chemicals), minimized secondary pollution and maximized materials value. In addition, the authors demonstrated high-performance asymmetric and symmetric systems based on multiple cell units. Overall, this work will be of great interest to the battery and energy storage community and should be considered for this journal. Before acceptance, however, several issues should be addressed:

- 1) The manuscript should provide details of disassembling the cell. Particularly, for the remaining metallic Na, which is highly active, should be taken care of. Large-scale adoption of Na metal in the recycling loop will cause safety problem. Similarly, hydrogen as a by-product may pose safety concern and should be considered for special treatment.
- 2) Some solutions for recycling electrolyte salt and solvent should be proposed in order to claim "fully recyclable battery" since every component should be considered. The recycling industrial has been struggling for years in dealing with organic electrolytes.
- 3) The direct regeneration process involves burning cathode slurry at 550C in the air, however, fluoridated binder (PVDF) cannot be directly burned due to their high toxicity to the environment and human being. A viable industry process should avoid such a route. In addition, a TAG curve showing the burning process could provide a detailed understanding of materials transformation during this treatment.
- 4) Direct regeneration has become a new direction for the battery recycling industry, and some important progress has been made recently. The introduction for the state-of-the-art should include more comprehensive literature survey for this part, three key references related to direct regeneration are recommended to include: J. Power Sources, 2017, 345, 78-84; ACS Energy Letters, 2018, 3, 1683-1692; Green Chemistry, 2018, 20, 851-862.

Minor typo issues:

Line 34: which "is".

Line 190: "involves"

Reviewer #2 (Remarks to the Author):

The upcoming e-mobility and needs for the energy storage are related to the substantial demands for the materials and metals needed for the battery production. Several materials used in the battery production exhibit criticality and the limited supply. Development of more effective and more environmentally friendly battery chemistries and designs is necessary for the future sustainability. In this manuscript, new cell design with more efficient recyclability has been proposed. In comparisons

to lithium-ion battery type lower amount of different materials was used to design the battery, which is highly needed in order to simplify the recycling. However lower amount of used materials does not have to necessary lead to the better or more sustainable design or recycling.

To discuss the proposed work reviewer comments are listed below:

There is a typo in the abstract – Herein we design...there is materials instead of materials.

I agree with the authors that recycling of lithium ion batteries has some disadvantages. However the recycling rates (especially for the hydrometallurgy) are high and moreover metals recovered and purified can be reuse in the new battery production supporting thus the sustainability and decreasing the demand for the mining.

Also the recycling of lithium ion batteries is not associated with the danger of exothermic reactions caused by the reaction of battery materials with the humidity or oxygen or with the hydrogen evolution (excluding the risk of self-ignition by accidents – which is handled by the mechanical pre-treatment).

Authors suggest that “highly toxic elemental V needs to be recycled”, what is the toxicity of NVP material. How is its level of toxicity compared to traditional NMC or NMA battery cathode materials? What is the risk level resulting from the exothermic reaction of $\text{Na(s)} + \text{H}_2\text{O (l)}$, since the ΔH of the reaction is negative (approximately -260 kJ) during the recycling process – if we consider a large scale recycling? Will it limit the scale of the possible recycling?

I do not agree with the authors that ventilation will be sufficient to decrease the risk of explosion during the recycling when hydrogen is generated (based on both proposed reactions). The limit for the hydrogen in the atmosphere (for the explosion) is very low. It is approximately 4%, which might be difficult to control in the large scale recycling and additional steps would be required to keep the hydrogen concentration low.

How feasible is supply of metallic Na for the larger scale production of NIB? How demanding is the process regarding the supply of raw materials and energy consumption?

How the use of Na contributes to the sustainability if the the recycling product is NaCl? Can Na be recycled for the production of new NIB?

What was the pH of the solution after the first reaction? Was NVP leaching observed? Was the leachate analyzed for vanadium or phosphor contamination (above pH 13, V can form VO_4^{3-} anion).

Authors mention that slurry containing NVP@C was sintered at 550°C in the presence of air and PVDF. Since there is a presence of carbon and oxygen it is expected that reductive atmosphere could be created by the presence of CO(g) at given temperature. Was the carbothermic reduction of NVP observed? If there was a partial decomposition of NVP, what was the portion of the NVP decomposed and what were the products?

Were the other products (off-gas or oil) analyzed as well? PVDF decomposes at temperature lower than 550°C and the products of its decomposition are not only HF, but also liquid products containing fluorine – which will mean another additional steps needed for the decontamination of the heat treatment products.

After the first step of the leaching with the water and NaOH, NVP slurry is recovered. Was it analyzed for the residual Na or Al contamination? Will the recovery of NVP require additional washing steps? How much of the waste water can be generated and how it can be re-use within the recycling process?

Was the washing solution (or leaching solution - NaOH) analyzed for the electrolyte residues? How would be contaminates treated or recovered?

How many times can be NVP reprocessed without losing the required properties? Was this studied in the current work?

What is the portion of NVP which can be re-use after the regeneration?

Current recycling of lithium-ion batteries is motivated mostly due to legislative regulations and price of the valuable metals (cobalt especially), which are recovered by the recyclers and then sold. What will be the motivation for the recycling of NIB if there is no high value component in it?

Sustainable recycling is the recycling where the recovered components can be reuse for the production

of new products. In the proposed work it is mostly NVP, which can be re-use without significant reprocessing. Steel needs to be re-melted and Al needs to be re-produced within the primary production – high energy demand.

More detailed analysis of the products and relevant suggestions for their treatment are needed to assess the sustainability of proposed battery and its recycling. Especially it needs to be specify, how many times NVP can be preprocessed without losing the properties.

Manuscript is not recommended for the publication, but it will be assessed after more detailed clarification is provided.

Reviewer #3 (Remarks to the Author):

This manuscript describes the design of a recyclable bipolar sodium ion battery for recyclability and sustainability of battery industry. The problems of battery waste and shortage of natural resources (lithium salt and cobalt mineral) are urgent and fundamental issues when batteries are used in a large-scale energy storage (such as electrical vehicles and smart grid). Nowadays, the recycling of the batteries are still not economically viable mainly because the current battery structures are not for recycling. The embedding of recycling mechanism into the battery is an innovative attempt in addressing this issue, which will attract extensive attentions from battery technologists. Hence, this work is extremely significant in the course of rapid development of battery technologies. The authors have successfully used the $\text{Na}_3\text{V}_2(\text{PO}_4)_3$ case to demonstrate this concept with detailed characterization and analysis. This work is well written and structured. I recommend this work for publication with minor changes.

1: There is a safety concern on the recycling process. Flammable H_2 gas is produced by the reaction of water with metallic sodium and slightly dilute basic aqueous solutions with aluminum. The authors should provide more information on lab safety and impact on the cost on this proposed technology.

2: The regeneration process of recycled $\text{Na}_3\text{V}_2(\text{PO}_4)_3$ is implemented at $800\text{ }^\circ\text{C}$ during 6 hours. Such a high temperature treatment is almost the same as the preparation of the pristine material, which is a high-energy consumption process. I would like the authors to comment the possibility to reduce the energy consumption in the recycling process.

3. In Fig. 2d, the data points are overlapping. The author should adjust the dot size to obtain a better contrast on the data sets.

4. The author should provide more discussion on the recycling of electrolyte, in terms of recycling mechanism and recycling cost.

Itemized Responses/Revisions to the original manuscript

Response to Referee #1:

General Comment: In this manuscript, Lu and Lin et al. reported a novel design concept of bipolar sodium-ion battery (SIB) for large-scale energy storage with the potential to significantly reduce the cost of electrode materials due to the viability of high-efficiency recycling. The usage of light-weight and low-cost Al current collect may also bring benefits, like further increase the system energy density, reducing the cell cost etc. Another important feature is the capability of direct regeneration of the NVP cathode material, which is significantly different from the conventional pyro- and hydrometallurgical recycling processes. The potential advantages of direct regeneration without fully destructing the active materials include reduced operation cost (process, energy, chemicals), minimized secondary pollution and maximized materials value. In addition, the authors demonstrated high-performance asymmetric and symmetric systems based on multiple cell units. Overall, this work will be of great interest to the battery and energy storage community and should be considered for this journal. Before acceptance, however, several issues should be addressed:

Author Reply: First of all, we really appreciate your positive comments on this work. We also thank your valuable comments and suggestions, which significantly help us to further improve the quality of our manuscript. According to your advices, some important data and corresponding discussion have been added in the revised manuscript. The responses for your specific comments are given below.

Comment 1-1. The manuscript should provide details of disassembling the cell. Particularly, for the remaining metallic Na, which is highly active, should be taken care of. Large-scale adoption of Na metal in the recycling loop will cause safety problem. Similarly, hydrogen as a by-product may pose safety concern and should be considered for special treatment.

Author Reply: Thanks for your valuable suggestion. In the revised manuscript, we provide more information of disassembling the coin-type NIB cell. In the video, the coin-type NIB cell is disassembled and indirectly put into the water. As for large-scale recycling NIB using metallic Na, the special treatment for the safety is necessary. Therefore, we supplement the discussion on safety concerns of metallic Na and by-product H₂.

Special treatment must be performed prior to battery recycling. As for the waste NIBs using metallic Na, remaining metallic Na is highly active. The waste NIB is first cut into small pieces. The resultant slags should be placed in a dry ventilated place. The Na can react with O₂ to form Na₂O (Equation 4) and subsequently partial Na₂O can react with CO₂ to form Na₂CO₃ (Equation 5). If this operation is executed well, the amount of by-product hydrogen significantly reduces with the addition of H₂O into the NIBs trash (Equation 6), compared with the direct water treatment.

The discussion is in Page 12 Line 7-12 in the revised manuscript.

Comment 1-2. Some solutions for recycling electrolyte salt and solvent should be proposed in order to claim “fully recyclable battery” since every component should be considered. The recycling industrial has been struggling for years in dealing with organic electrolytes.

Author Reply: Thanks for your significant reminding. As your comments, the electrolyte salt and solvents are difficult to recycle. It is attributed to a small amount of organic electrolyte used in the battery. The cost proportion of the electrolyte is only *ca.* 15% of the battery cost (*Nat. Rev. Mater.*, 2018, **3**, 18013). Therefore, there is less economic and research interest in recycling electrolyte.

The recovery of the electrolyte can share with the LIB technologies. In general, electrolyte salts are dissolved in organic solvents (e.g. propylene carbonate and ethylene

carbonate) with functional additives (e.g. vinyl carbonate and biphenyl). Currently, the electrolyte recycling methods are mainly classified as vacuum pyrolysis, organic solvent extraction, and supercritical CO₂ extraction. (J. Hazard. Mater., 2011, 194, 378-384) Referring to work by Liu et al., high-efficiency recycling of electrolyte has been demonstrated and the recycled electrolyte can be reused in new batteries. (Phys. Chem. C 2017, 121, 4181–4187; RSC Adv., 2014, 4, 54525-54531; Int. J. Electrochem. Sci., 2016, 11 7594-7604).

The discussion is in Page 11 Line 16-21 in the revised manuscript.

Comment 1-3. The direct regeneration process involves burning cathode slurry at 550C in the air, however, fluoridated binder (PVDF) cannot be directly burned due to their high toxicity to the environment and human being. A viable industry process should avoid such a route. In addition, a TAG curve showing the burning process could provide a detailed understanding of materials transformation during this treatment.

Author Reply: Thanks for your valuable reminding. As for the pollution of off-gas from the burn of PVDF, we consulted the environmental experts. The fluoridated binder (PVDF) is first directly burned in order to remove the binder from the recycled NVP@C slurry. However, this process can produce volatile organic compounds (VOCs), which are of high toxicity to the environment and human being. The off-gas must be treated further at higher temperature, enabling efficient decontamination of VOCs. (*Studies in Environmental Science*, 1994, 61, 263-276)

The discussion is in Page 12 Line 17-20 in the revised manuscript.

We also provide more discussions on TGA curve of NVP@C. The TGA curve of the NVP@C materials shows carbon coating on the surface of NVP particles is burned at the temperature of 500 °C. As the temperature increases, the NVP is oxidized by the value of V from +3 to +5.

The discussion is in Page 8 Line 1-4 in the revised manuscript.

Comment 1-4. Direct regeneration has become a new direction for the battery recycling industry, and some important progress has been made recently. The introduction for

the state-of-the-art should include more comprehensive literature survey for this part, three key references related to direct regeneration are recommended to include: J. Power Sources, 2017, 345, 78-84; ACS Energy Letters, 2018, 3, 1683-1692; Green Chemistry, 2018, 20, 851-862.

Author Reply: Thanks for your valuable reminding. We have cited these literatures in the revised manuscript.

Comment 1-5. Minor typo issues:

Line 34: which “is”.

Line 190: “involves”

Author Reply: Thanks for your valuable reminding. We have corrected the confusion and mistakes in the revised manuscript.

Response to Referee #2:

General Comment: The upcoming e-mobility and needs for the energy storage are related to the substantial demands for the materials and metals needed for the battery production. Several materials used in the battery production exhibit criticality and the limited supply. Development of more effective and more environmentally friendly battery chemistries and designs is necessary for the future sustainability. In this manuscript, new cell design with more efficient recyclability has been proposed. In comparisons to lithium-ion battery type lower amount of different materials was used to design the battery, which is highly needed in order to simplify the recycling. However lower amount of used materials does not have to necessary lead to the better or more sustainable design or recycling. To discuss the proposed work reviewer comments are listed below:

Author Reply: Firstly, we sincerely appreciate for the referee's approval of battery recyclability and sustainability. To address this issue, we propose the concept of early recyclable battery design to facilitate the post battery recycling. In this work, we design bipolar electrode structure for sodium ion battery (NIB), and demonstrate a recyclable NIB, in order to address the concerns on the resource reserve and environmental pollution of end-of-life batteries simultaneously. We also are grateful for your valuable comments and suggestions, which significantly help us to further improve the quality of our manuscript. According to your advice, some important data and corresponding discussion have been added in the revised manuscript. The responses for your specific comments are given below.

Comment 2-1. There is a typo in the abstract – Herein we design...there is materials instead of materials.

Author Reply: Thanks for your kind reminding. We have corrected this typo in the revised manuscript.

Comment 2-2. I agree with the authors that recycling of lithium ion batteries has some disadvantages. However the recycling rates (especially for the hydrometallurgy) are high and moreover metals recovered and purified can be reuse in the new battery production supporting thus the sustainability and decreasing the demand for the mining. Also the recycling of lithium ion batteries is not associated with the danger of exothermic reactions caused by the reaction of battery materials with the humidity or oxygen or with the hydrogen evolution (excluding the risk of self-ignition by accidents – which is handled by the mechanical pre-treatment).

Author Reply: Thanks for your information on hydrometallurgy process. Although the hydrometallurgy process has high efficiency for metals recyclability and purification, the main concern focuses on using massive chemical agents, which lead to high cost and barely make economic sense. Therefore, we attempt to decrease the cost in recycling, achieving sufficient benefit from end-of-life batteries. According to reports on the recycling of lithium ion batteries, our recycling operation in NIBs is very safe and controlled. Therefore, this experience from recycling sodium ion batteries can be greatly shared with lithium ion batteries in the future. We have also corrected the statement on recycling rates of hydro- or pyrometallurgy process.

The discussion is in Page 2 Line 14-19 in the revised manuscript.

Comment 2-3. Authors suggest that “highly toxic elemental V needs to be recycled”, what is the toxicity of NVP material. How is its level of toxicity compared to traditional NMC or NMA battery cathode materials?

Author Reply: Thanks for your helpful reminding. We have cited the reference of “Processing of vanadium,” expressing “Vanadium and its many compounds are toxic and require careful handling.”

In addition, the toxicity of metals mainly depends on their value. For example, the metallic vanadium is low toxicity, however, the oxide and some other salts of vanadium have high toxicity, especially for its value of +5. The NVP exhibits the V^{5+} at the charging state and V^{3+} at discharging state. The values of Ni, Mn, Co, Al in NMC or

NMA materials have +2, and +3. We check the essential information of materials from the Sigma-Aldrich company.

1. The signal word for V^{3+} is warning.

(https://www.sigmaaldrich.com/catalog/product/aldrich/463744?lang=en®ion=AU&utm_medium=referral&utm_source=pubchem&utm_campaign=pubchem_2017)

2. The signal word for V^{5+} is danger.

(https://www.sigmaaldrich.com/catalog/product/aldrich/204854?lang=en®ion=AU&utm_medium=referral&utm_source=pubchem&utm_campaign=pubchem_2017)

3. The signal word for NMC is warning.

(<https://www.sigmaaldrich.com/MSDS/MSDS/DisplayMSDSPage.do?country=AU&language=en&productNumber=761001&brand=ALDRICH&PageToGoToURL=https%3A%2F%2Fwww.sigmaaldrich.com%2Fcatalog%2Fsearch%3Fterm%3D346417-97-8%26interface%3DCAS%2520No.%26N%3D0%2B%26mode%3Dpartialmax%26lang%3Den%26region%3DAU%26focus%3Dproduct>)

4. The signal word for $Co^{2+/3+}$ is danger.

(https://www.sigmaaldrich.com/catalog/product/aldrich/203114?lang=en®ion=AU&utm_medium=referral&utm_source=pubchem&utm_campaign=pubchem_2017)

5. The signal word for Al^{3+} is none.

(<https://www.sigmaaldrich.com/MSDS/MSDS/DisplayMSDSPage.do?country=AU&language=en&productNumber=642991&brand=ALDRICH&PageToGoToURL=https%3A%2F%2Fwww.sigmaaldrich.com%2Fcatalog%2Fproduct%2Faldrich%2F642991%3Flang%3Den>)

The toxicity of Co is much lower than that of Al. The NMA uses Al instead of Co and thus the toxicity of NMA is low than that of NMC.

In sum, the toxicity of NVP material is higher than that of NMC and NMA.

Comment 2-4. What is the risk level resulting from the exothermic reaction of $Na(s)+H_2O(l)$, since the ΔH of the reaction is negative (approximately -260 kJ)

during the recycling process – if we consider a large scale recycling? Will it limit the scale of the possible recycling?

Author Reply: Thanks for your useful suggestion. The risk level resulting from the exothermic reaction of Na(s)+H₂O(l) is high, especially at large scale recycling process. Massive hydrogen is produced during the recycling process, leading to the safety concerns. Therefore, large-scale recycling of NIB using metallic Na anode is barely carried out.

To this end, on one hand, we propose an alternative strategy-----symmetrical NIB configuration using NVP as both the cathode and anode. The diluted NaOH solution instead of water was used to separate NVP and recycle elemental Al in the recycling process. Because both the cathode and anode use identical NVP@C material, the recovery for this symmetric NIB cell is therefore as convenient as the asymmetric NIB cell mentioned-above. The solid-component recycling efficiency of ca. 99.7% is achieved with this system. No usage of metallic Na in bipolar significantly reduces the amount of ΔH in recycling process. (Please see Page 8 Line 15-19 in the revised manuscripte)

On the other hand, special treatment must be performed prior to battery recycling. As for the waste NIBs using metallic Na, remaining metallic Na is highly active. The waste NIB is first cut into small pieces. The resultant slags should be placed in a dry ventilated place. The Na can react with O₂ to form Na₂O (Equation 4) and subsequently partial Na₂O can react with CO₂ to form Na₂CO₃ (Equation 5). If this operation is executed well, the amount of by-product hydrogen significantly reduces with the addition of H₂O into the NIBs trash (Equation 6), compared with the direct water treatment. In addition, the ventilation must be carried out from beginning to end in battery recycling process.

The discussion is in Page 12 Line 7-12 in the revised manuscript.

Comment 2-5. I do not agree with the authors that ventilation will be sufficient to decrease the risk of explosion during the recycling when hydrogen is generated (based on both proposed reactions). The limit for the hydrogen in the atmosphere (for the explosion) is very low. It is approximately 4%, which might be difficult to control in the large scale recycling and additional steps would be required to keep the hydrogen concentration low.

Author Reply: We really thank for treasured reminding of the safety recycling. In addition to ventilation for decreasing the risk of hydrogen explosion, during recycling we attempt to provide more possible solutions to address safety problems by reducing the amount of hydrogen as by-product. Please refer to the strategies illustrated in Comment 2-4.

In addition, Al foil can be collected prior to full dissolution. We found that the surface of Al foil is readily corroded by an alkali solution, leading to rapid separation of electrode slurry and Al foil, as shown in Figure R2-1. Therefore, fast recycling of Al foil delaminated from the electrode can reduce the amount of H₂ in the recycling process.

Figure R2-1 The separation of electrode slurry and Al foil in recycling process.

This Figure R2-1 has been updated as Figure S7 in the supporting information and corresponding discussion is in Page 12 Line 13-15 in the revised manuscript.

Comment 2-6. How feasible is supply of metallic Na for the larger scale production of NIB? How demanding is the process regarding the supply of raw materials and energy consumption?

Author Reply: Thanks for your helpful reminding. In general, the metallic Na is mainly produced by the electrolysis of molten salts. Supplies of sodium salts are highly available. The United States alone produces 23 billion tons of soda ash, which is far more than global yield of lithium salts (*Chem. Soc. Rev.*, 2017, 46, 3529). In addition, one good example is high-temperature Na-Sulfur batteries (HT-NSBs) technology that were successfully commercialized in the 1980s. The NGK Company has installed the HT-NSBs at nearly 200 locations worldwide to deliver a cumulative installation base of 3700 MWh. Therefore, the supply of metallic Na for larger-scale production of NIB is highly feasible.

The corresponding discussion is in Page 3 Line 4-10 in the revised manuscript.

Comment 2-7. How the use of Na contributes to the sustainability if the recycling product is NaCl? Can Na be recycled for the production of new NIB?

Author Reply: Thanks for your valuable suggestion. The recycling of metallic anode is also highly feasible. Currently, metallic Na and Al are mainly produced by the electrolysis of molten salts at high temperature. We can expect that metallic Na and Al can be refined well by melting recycled NaCl and Al(OH)₃ at 600 and 950 °C for further usage, respectively. The discussion is in Page 12 Line 2-5 in the revised manuscript.

Comment 2-8. What was the pH of the solution after the first reaction? Was NVP leaching observed? Was the leachate analyzed for vanadium or phosphor contamination (above pH 13, V can form VO₄⁻³ anion).

Author Reply: Thanks for your kind reminding. The pH of the solution is 11.5. Subsequently, the leachate was tested by the ICP measurement. The results show that vanadium or phosphate concentration is less than 2 ppb in the leachate, indicating that the NVP is very stable in basic solution.

The discussion is in Page 8 Line 7-9 in the revised manuscript.

Comment 2-9. Authors mention that slurry containing NVP@C was sintered at 550°C in the presence of air and PVDF. Since there is a presence of carbon and oxygen it is expected that reductive atmosphere could be created by the presence of CO(g) at given temperature. Was the carbothermic reduction of NVP observed? If there was a partial decomposition of NVP, what was the portion of the NVP decomposed and what were the products?

Author Reply: Thanks for your kind reminding. To our knowledge, the formation of CO (g) often occurs insufficient oxygen atmosphere. Meanwhile, carbothermic reduction of NVP mainly occurs in the inert atmosphere. However, in this work, the thermal treatment of NVP@C slurry is in the air where oxygen is sufficient. According to the TGA results (Figure R2-2a), there is an increase in weight of NVP sample, indicating that the NVP was oxidized at 550 °C. (Please see the Page 8 Line 1-4)

Also there is no decomposition of NVP sample, according to the supplementary XRD test (Figure R2-2b). The XRD patterns (Fig. 4a) are well indexed to high-crystallinity NASICON structure (JCPDS No. 62-0345) without any impurity. (Please see the Page 10 Line 9-11)

Figure R2-2 (a) TGA curves of the NVP@C; (b) XRD spectra of reprocessed and original NVP@C.

Comment 2-10. Were the other products (off-gas or oil) analyzed as well? PVDF decomposes at temperature lower than 550°C and the products of its decomposition are not only HF, but also liquid products containing fluorine – which will mean another additional steps needed for the decontamination of the heat treatment products.

Author Reply: Thanks for your kind reminding. The waste gas should be considered for the safety issue of environment and human being. The fluoridated binder (PVDF) is first directly burned in order to remove the binder from recycled NVP@C slurry. However, this process can produce volatile organic compounds (VOCs), which are of high toxicity to environment and human being. The off-gas must be treated further at higher temperature, enabling efficient decontamination of VOCs. (*Studies in Environmental Science, 1994, 61, 263-276*)

The discussion is in Page 12 Line 17-20 in the revised manuscript.

Comment 2-11. After the first step of the leaching with the water and NaOH, NVP slurry is recovered. Was it analyzed for the residual Na or Al contamination? Will the recovery of NVP require additional washing steps? How much of the waste water can be generated and how it can be re-use within the recycling process?

Author Reply: Thanks for your kind reminding. In recycled process, the NVP slurry was recycled by filtration and then an additional washing steps for recycled NVP slurry was performed, leading to a 20% increase in waste water. These waste water from additional washing steps can be reused in the recycling tank.

The resultant NVP slurry recycled from waste NIBs was tested by the ICP measurement. The results show that Al residuals was not detected in recycled NVP sample.

The discussion is in Page 12-13 in the revised manuscript.

Comment 2-12. Was the washing solution (or leaching solution - NaOH) analyzed for the electrolyte residues? How would be contaminates treated or recovered?

Author Reply: Thanks for your valuable advice. The electrolyte was difficult to recycle due to a small amount of organic electrolyte used in coin-type NIB cell.

Therefore, the leaching solution has the electrolyte residues. In the scope of this work, we focus on solid-component recycling in spent NIB with bipolar electrode structure.

The recovery of the electrolyte can share with the LIB technologies. In general, electrolyte salts are dissolved in organic solvents (e.g. propylene carbonate and ethylene carbonate) with functional additives (e.g. vinyl carbonate and biphenyl). Currently, the electrolyte recycling methods are mainly classified as vacuum pyrolysis, organic solvent extraction, and supercritical CO₂ extraction. (J. Hazard. Mater., 2011, 194, 378-384) Referring to work by Liu et al., high-efficiency recycling of electrolyte has been demonstrated and the recycled electrolyte can be reused in new batteries. (Phys. Chem. C 2017, 121, 4181-4187; RSC Adv., 2014, 4, 54525-54531; Int. J. Electrochem. Sci., 2016, 11 7594-7604).

The discussion is in Page 11 Line 16-21 in the revised manuscript.

Comment 2-13. How many times can be NVP reprocessed without losing the required properties? Was this studied in the current work?

Author Reply: Thanks for your kind reminding. The NVP recycling from tested NIB cells was repeatedly implemented again. The reprocessed NVP still exhibit very good performance with stable long-term cycling in the NIBs, as shown in Figure R2-3.

Figure R2-3 The NVP@C composite at secondary recycling. (a) XRD spectrum; (b) The charge and discharge curves at 5 C; (c) Cycle performance with 1000 cycles at the rate of 5 C.

We only reprocessed NVP for two times in this work. During this period, the recycled NVP demonstrate very good performance with stable long-term cycling in the full NIB. We believe that the recycled NVP can be further processed for many times; however, due to timeframe, we think the current work has demonstrated the ability of close-loop recycling of NVP.

Figure R2-3 has been updated as Figure S16 in the supporting information and the corresponding discussion is in Page 10 Line 8-10 in the revised manuscript.

Comment 2-14. What is the portion of NVP which can be re-use after the regeneration?

Author Reply: Thanks for your kind reminding. We re-used 100% NVP in the new NIB. This statement have been added in experiment.

Comment 2-15. Current recycling of lithium-ion batteries is motivated mostly due to legislative regulations and price of the valuable metals (cobalt especially), which are recovered by the recyclers and then sold. What will be the motivation for the recycling of NIB if there is no high value component in it?

Author Reply: Thanks for your significant suggestion. As shown in Figure R2-3, the cathode materials account for 30-40% of battery cost (*Nat. Rev. Mater.*, 2018, 3, 18013). Currently, the initiative for the recycling waste batteries is mainly from economic interests. Compared with the economic motivation, the environmental awareness will be the new motivation for the recycling of NIB. The NIBs are considered as the promising candidate for large-scale energy storage, calling for extremely large battery production and consequently waste batteries. Developing recyclable technologies of metal resources is the most cost-effective way by avoiding repeated metallurgical process, reducing the emissions of greenhouse gas. More importantly, the lessons from White Plastics have taught us that prescient product design could help to increase recycling and reduce the environmental impact.

Figure R2-4 A cost and resource analysis of lithium and sodium-ion batteries. The statement is added in Page 3 Line 9-11 in the revised manuscript.

Comment 2-16. Sustainable recycling is the recycling where the recovered components can be reuse for the production of new products. In the proposed work it is mostly NVP, which can be re-use without significant reprocessing. Steel needs to be re-melted and Al needs to be re-produced within the primary production – high energy demand.

Author Reply: Thanks for your treasured reminding. The NVP is directly regeneration without complex synthese process. Compared with NVP, the reuse of steel and Al sources needs to be melted with high energy demand. However, this process is unavoidable and beyond the scope of this work. In this work, we have provided more information about reducing materials and energy consumption.

Unlike unipolar electrode structure, bipolar electrode structure enables the rechargeable battery with lower amount of used steel and Al, which favor to decrease energy consumption.

The statement is added in Page 11 Line 5-9 in the revised manuscript.

Comment 2-17. More detailed analysis of the products and relevant suggestions for their treatment are needed to assess the sustainability of proposed battery and its recycling. Especially it needs to be specify, how many times NVP can be preprocessed without losing the properties. Manuscript is not recommended for the publication, but it will be assessed after more detailed clarification is provided.

Author Reply: Thanks for all your suggestion and comments, which significantly help us to further improve the quality of our manuscript. The new data and corresponding discussions have been added in the revised manuscript. In this work, we have reprocessed NVP two times without losing the properties, indicating the ability of close-loop recycling of NVP. Although the effectiveness of battery recycling was only demonstrated with two times, its concept as a new battery design concept can significantly benefit the generic battery system, enabling recyclability and sustainability of a battery. We believe future iteration of this concept will lead to the novel battery design and electrode materials reshape battery structure in both academic/industrial settings. We sincerely hope that the revisions can meet your requirements and make the revised manuscript more suitable to be published in *Nature Communications*.

Response to Referee #3:

General Comment: This manuscript describes the design of a recyclable bipolar sodium ion battery for recyclability and sustainability of battery industry. The problems of battery waste and shortage of natural resources (lithium salt and cobalt mineral) are urgent and fundamental issues when batteries are used in a large-scale energy storage (such as electrical vehicles and smart grid). Nowadays, the recycling of the batteries are still not economically viable mainly because the current battery structures are not for recycling. The embedding of recycling mechanism into the battery is an innovative attempt in addressing this issue, which will attract extensive attentions from battery technologists. Hence, this work is extremely significant in the course of rapid development of battery technologies. The authors have successfully used the $\text{Na}_3\text{V}_2(\text{PO}_4)_3$ case to demonstrate this concept with detailed characterization and analysis. This work is well written and structured. I recommend this work for publication with minor changes.

Author Reply: Thank you so much for your positive comments. We sincerely appreciate for the referee's valuable comments and suggestions, which are undoubtedly beneficial to improve the quality of our manuscript. According to your helpful advices, some necessary data and corresponding discussion have been added in the revised manuscript. The responses for your specific comments are listed below.

Comment 3-1. There is a safety concern on the recycling process. Flammable H_2 gas is produced by the reaction of water with metallic sodium and slightly dilute basic aqueous solutions with aluminum. The authors should provide more information on lab safety and impact on the cost on this proposed technology.

Author Reply: Thanks for your valuable suggestion. The waste NIB is first cut into small pieces. The metallic Na is exposed to the air. The most of metallic Na will react with O_2 in the air to form Na_2O (Equation 4) since the metallic Na is very active. The subsequent reaction is that the partial Na_2O react with CO_2 to form Na_2CO_3 (Equation 5).

With the H₂O added into the trash of NIBs, the amount of H₂ as by-product is much less (Equation 6), compared with the direct water treatment.

The discussion is in Page 12 Line 7-12 in the revised manuscript.

Comment 3-2. The regeneration process of recycled Na₃V₂(PO₄)₃ is implemented at 800 °C during 6 hours. Such a high temperature treatment is almost the same as the preparation of the pristine material, which is a high-energy consumption process. I would like the authors to comment the possibility to reduce the energy consumption in the recycling process.

Author Reply: Thanks for your valuable suggestion. In this work, we emphasize the highly effective recycling of electrode units, which are the most valuable battery components. Unlike the traditional recycling battery process, this proposed strategy of battery design is energy-saving and low cost. Because the new NVP@C is obtained by only reprocessing recycled NVP materials rather than using NVP precursor. In addition, the thermal treatment for recycled NVP is time-saving compared with NVP precursor. Therefore, both the energy consumption and materials cost is significantly reduced.

Comment 3-3. In Fig. 2d, the data points are overlapping. The author should adjust the dot size to obtain a better contrast on the data sets.

Author Reply: Thanks for your kind reminding. We have adjusted the curves to make the results clear.

Comment 3-4. The author should provide more discussion on the recycling of electrolyte, in terms of recycling mechanism and recycling cost.

Author Reply: Thanks for your valuable suggestion. The recovery of the electrolyte can share with the LIB technologies. In general, electrolyte salts are dissolved in organic solvents (e.g. propylene carbonate and ethylene carbonate) with functional additives (e.g. vinyl carbonate and biphenyl). Currently, the electrolyte recycling methods are mainly classified as vacuum pyrolysis, organic solvent extraction, and supercritical CO₂ extraction. (J. Hazard. Mater., 2011, 194, 378-384) Referring to work by Liu et al., high-efficiency recycling of electrolyte has been demonstrated and the recycled electrolyte can be reused in new batteries. (Phys. Chem. C 2017, 121, 4181–4187; RSC Adv., 2014, 4, 54525-54531; Int. J. Electrochem. Sci., 2016, 11 7594-7604).

The discussion is in Page 11 Line 16-21 in the revised manuscript.

List of Revisions

Revisions of spelling Errors/Typos:

1. P1: "mateials" → "materials";
2. P10: " involves" → "involve";

Major revision and additions of sentences:

1. P2: Hydro- and pyro-metallurgical recovery for valuable metals in the LIB has been widely used,..... and substantial production of non-ecologically friendly solid, liquid, and gas wastes.
2. P3: One good example is high-temperature Na-Sulfur batteries (HT-NSBs) that were successfully.....considered as a safer energy storage technology compared with the HT-NSBs.
3. P3: Historically, learned from the crisis of..... recycling and reduce the environmental impact.
4. P8-9: An additional washing steps for recycled NVP slurry was performed, leading to a 20% increase in waste water..... Therefore, the obtained cathode slurry was sintered in the air at 550 oC to recycle NVP.
5. P11: Fourth, the NVP is directly regeneration without complex synthese process and.....which favor to decrease energy consumption in recycling.
6. P11-13: Further explanation is needed here to claim our fully recyclable NIB. Firstly, the recovery of the electrolyte can share with the LIB technologies.....The off-gas must be treated further at higher temperature, enabling efficient decontamination of VOCs.
7. P16: The recycling, reprocessing and reuse of the NVP@C materials in bipolar NIBs is performed two times. In each time, all NVP@C is reused in the NIBs after the regeneration.

Revision and Additions of Figures/Tables:

1. Figure S7;
2. Figure S15;
3. Figure S16.

REVIEWERS' COMMENTS:

Reviewer #1 (Remarks to the Author):

The revised version has addressed all of my questions and concerns. It can be accepted in this form.

Reviewer #2 (Remarks to the Author):

No further comments.

Reviewer #3 (Remarks to the Author):

The new manuscript has been well revised according to the reviewers' comments. And in its current form, the revised manuscript with a professional presentation and editing can be regarded as a full rigorous and scientific paper. Therefore, I suggest that this work is now suitable for publication.